# Role of Leu72Met of *GHRL* and Gln223Arg of *LEPR* Variants on Food Intake, Subjective Appetite, and Hunger-Satiety Hormones

**DOI:** 10.3390/nu14102100

**Published:** 2022-05-18

**Authors:** Tania Sanchez-Murguia, Nathaly Torres-Castillo, Lisset Magaña-de la Vega, Saraí Citlalic Rodríguez-Reyes, Wendy Campos-Pérez, Erika Martínez-López

**Affiliations:** 1Instituto de Nutrigenética y Nutrigenómica Traslacional, Centro Universitario de Ciencias de la Salud, Universidad de Guadalajara, Guadalajara 44100, JA, Mexico; ln.taniasanchez@gmail.com (T.S.-M.); nathaly.torrescas@academicos.udg.mx (N.T.-C.); lissma14@gmail.com (L.M.-d.l.V.); citlalic.rodriguez@academicos.udg.mx (S.C.R.-R.); wendy_yareni91@hotmail.com (W.C.-P.); 2Doctorado en Ciencias de la Nutrición Traslacional, Centro Universitario de Ciencias de la Salud, Universidad de Guadalajara, Guadalajara 44100, JA, Mexico; 3Departamento de Biología Molecular y Genómica, Centro Universitario de Ciencias de la Salud, Universidad de Guadalajara, Guadalajara 44100, JA, Mexico

**Keywords:** leptin, ghrelin, appetite, carbohydrate intake, hunger, satiety

## Abstract

Appetite regulation has been recognized as a promising target for the prevention of obesity, which has become a worldwide health issue. Polymorphisms in the genes of hormones or receptors including Leu72Met for *ghrelin* and Gln223Arg for the *leptin receptor* could play a role in dietary intake, hunger, and satiety process. The aim of this study was to analyze subjective appetite assessments, dietary intake, and appetite hormones in relationship to these polymorphisms. Subjects (*n* = 132) with normal BMIs were enrolled. Dietary intake was analyzed with 3-day diet records. Subjective appetite was measured by visual analogue scales. Biochemical parameters were measured after 12 h of fasting and 120′ following ingestion of a test meal. Ghrelin and leptin levels were measured by ELISA assay (enzyme-linked immunosorbent assay) and insulin by chemiluminescence assay. The polymorphisms were determined by allelic discrimination using TaqMan^®^ probes. Fasting ghrelin levels differed significantly between men and women. The consumption of fruit and bread/starch with added sugar servings, as indicated by dietary records, and measured ghrelin levels were higher in carriers of Leu72Met/Met72Met compared to Leu72Leu carriers; total sugar intake was higher in Gln223Gln carriers than in Gln223Arg/Arg223Arg carriers. In conclusion, the Leu72Met and Gln223Arg polymorphism in *ghrelin* and *LEPR* may contribute to differential responses to a standardized meal as evidenced by higher postprandial levels of ghrelin and may also contribute to a higher dietary sugar intake.

## 1. Introduction

Obesity is a worldwide health issue that has proven to be hard to prevent and treat. A leading cause of obesity is the absence of energy homeostasis characterized by an imbalance between the intake and expenditure of energy, which depends on mechanisms that modulate the amount of food consumed and the frequency of meals [1,2]. Therefore, the regulation of appetite has been recognized as a promising target to prevent obesity.

Appetite can be evaluated through visual analogue scales (VAS) for aspects of appetite such as hunger, fullness, and desire to eat. In addition, the quantification of peptides that influence gastrointestinal motility, satiety, and hunger provide a biochemical insight into appetite [3]. A key peripheral peptide involved in hunger is ghrelin, whereas the hormones leptin and insulin regulate the satiety process and participate in the control of energy intake [4]. 

Ghrelin is a 28-aminoacid peptide produced and secreted by the P/D1 type cells of the stomach. Upon interaction with its receptor, signals are sent to the ventral tegmental area (VTA) to stimulate food-seeking behavior and the ingestion of palatable food [5]. In turn, leptin is mainly secreted by the adipose tissue and binds to leptin receptor (LEPR) on the neurons of the hypothalamus, especially in the arcuate nucleus (ARC) [6]. Activation of leptin signaling in the hypothalamus increases neuronal activity, resulting in reduced food intake and increased energy expenditure [7]. Lastly, insulin enters the brain via saturable transport across the blood–brain barrier (BBB) and binds to insulin receptors (IR) also located in the hypothalamus [8]. Its effects include the modulation of afferent and efferent signaling pathways that induce a decrease in energy intake, diminishment of motivation, and hedonic aspects caused by food consumption [9,10].

In addition to environmental and physiological responses, some genetic factors may be related to an imbalance in appetite control and food intake, which can increase the susceptibility to being overweight or obese [11,12]. The genetic background includes some single nucleotide polymorphisms (SNPs) in genes that code for proteins involved in the hypothalamic control of food intake and energy balance, or genetic variants in genes that code for gastrointestinal peptides involved in hunger and satiety [13]. 

The gene *ghrelin (GHRL)*, which codes for ghrelin, contains the rs696217 polymorphism (Leu72Met) that has been associated with early onset of obesity [14], higher risk of binge-eating behavior [15], alcohol consumption [16], and dietary fat intake [17]. However, other authors did not find associations between the (Leu72Met) polymorphism and differences in weight loss [18], dietary intake [19], or eating disorders such as anorexia or bulimia [20]. On the other hand, the study of the rs1137101 (Gln223Arg) polymorphism in the leptin receptor gene (*LEPR*) has been focused mainly on obesity risk with controversial results [21,22,23,24], wherein a small number of studies related it to appetite or food intake [24]. 

Hence, there is a relative paucity of studies describing how these two genetic variants affect appetite responses. Therefore, this study aimed to analyze subjective appetite, dietary intake, and appetite hormones according to the *GHRL* (Leu72Met) and *LEPR* (Gln223Arg) polymorphisms.

## 2. Materials and Methods

### 2.1. Study Subjects 

In this quasi-experimental study design (which aimed to evaluate the response of a meal and did not use randomization or include a control group) [25], 132 unrelated adults from western Mexico were included. The participants were recruited using posters and flyers from 2019 to 2020. The study was carried out at the Institute of Translational Nutrigenetics and Nutrigenomics of the University of Guadalajara. The sample size was calculated considering a statistical power of 80% and α = 0.05. Inclusion criteria for subjects included age between 18 and 25 years, normal weight according to body mass index (BMI) classification (18.5–24.9 kg/m^2^) and having the habit of breakfast. Exclusion criteria were subjects who were vegans or vegetarians, with food allergies, elite-athletes, medical or in nutritional treatment for weight loss, appetite-altering medications, smokers, having respiratory symptoms, aversion to breakfast ingredients, and women who were pregnant, breastfeeding, or using hormonal contraceptives. Finally, only subjects who signed a written informed consent and complete all the sessions were included (Figure 1). This study was approved by the Ethics and Biosafety Committees for Human Research of the University of Guadalajara (registration number CI-03619) and all procedures were performed according to the Declaration of Helsinki (World Medical Association 2013). 

### 2.2. Procedure

The subjects were summoned twice in a state of fasting for 12 h. In the first session, we obtained the anthropometric measurements and gathered the clinical history. In the second session, we took vital signs and a peripheral blood sample, and asked them to fill a VAS. Then, they consumed an isocaloric breakfast (see breakfast design section). At the end of breakfast, postprandial VAS at 30’, 60’, 90’, and 120’ were applied. Finally, a blood sample was taken at 120’ postprandial. Details are shown in Figure 2.

### 2.3. Anthropometric Measurements

Anthropometric measurements were taken after 12 h of fasting in all participants. Measurements were performed in light clothes and without shoes. Height was determined using a stadiometer with a precision of 0.1 cm and a measuring range up to 205 cm (SECA^®^ stadiometer, SECA GMBH & Co., Hamburg, Germany; model 213). Body composition was analyzed by electrical bioimpedance (Inbody 370, Biospace Co., Seoul, Korea, 250 kg capacity, 0.1 kg precision). Waist circumference was measured in the narrowest diameter between the last rib and the iliac crest using a Lufkin Rosscraft^®^ tape (Lufkin Rosscraft^®^ metal tape measure, Houston, Texas, USA; model W606, range 0 to 200 cm, accurate to 0.1 cm). Blood pressure was measured with an Omron Automatic arm digital blood pressure monitor (HEM-7130 Omron Healthcare Co., Ltd., Kyoto, Japan) after 15 min of rest, for which subjects were instructed to sit with their backs touching the back of the chair, to rest their arms on a horizontal surface, and to keep their legs without crossing. 

### 2.4. Biochemical Determinations

Peripheral blood samples were taken by a venous puncture after 12 h of fasting and at 120 min postprandial, and they were immediately centrifuged at 3500 rpm to obtain serum. The serum was separated and stored at −80 °C for later use. The concentration of glucose, triglycerides, total cholesterol, and high-density lipoprotein cholesterol (HDL-c) cholesterol was measured with a dry chemistry analyzer Vitros 350 Chemistry (Ortho-Clinical Diagnostics, Johnson & Johnson Services Inc., Rochester, NY, USA). Low-density lipoprotein cholesterol (LDL-c) was calculated with the Friedewald formula, except when triglycerides levels were higher than 400 mg/dL [26]. Very-low-density lipoprotein cholesterol (VLDL-c) was calculated as total cholesterol minus the sum of LDL-c + HDL-c.

### 2.5. Quantification of Appetite Hormones

Ghrelin and leptin levels were measured by ELISA assay (Enzyme-Linked Immuno-Sorbent Assay) according to the manufacturer’s recommendations. The serum ghrelin levels were determined using RayBio^®^, model Human GHRL/Ghrelin ELISA Kit catalog number ELH-GHRL (RayBiotech, Norcross, GA, USA). The serum leptin levels were determined using ALPCO^®^ Leptin ELISA (Ultrasensitive) catalog number 22-LEPHUU-E01 (Keewaydin Drive, Salem, MA, USA). The serum insulin levels were determined by an assay based on the chemiluminescence (CLIA) principle in a LIAISON^®^ analyzer according to the manufacturer’s recommendations using the LIAISON^®^ Insulin kit (REF 310360) DiaSorin S.p.A (Via Crescentino snc-1340 Saluggia (VC) Italy). 

### 2.6. DNA Extraction and Genotyping

We used 200 µL of peripheral blood sample for the extraction of genomic deoxyribonucleic acid (gDNA). The assay was performed with the High Pure Polymerase Chain Reaction (PCR) Template Preparation Kit (Roche Diagnostics, Indianapolis, IN, USA). The quantification and purity of gDNA were analyzed with the Multiskan™ SkyHigh Microplate Spectrophotometer (Thermo Fisher Scientific Inc., Singapore). Genotyping of the Leu72Met (rs696217) of the *GHRL* gene and Gln223Arg (rs1137101) of the *LEPR* gene was performed by allelic discrimination using TaqMan^®^ probes. (Drug Metabolism Assay, Applied Biosystems, Foster City, CA, USA). The amplification reaction was carried out in a Light Cycler^®^ 96 Real-Time PCR System (Roche Diagnostics, Mannheim, Germany). The amplification protocol consisted of a pre-incubation stage at 95 °C for 10 min, then 40 cycles of 15 s each at 95 °C, and 1 min at 60 °C. A 10 μL mixture was prepared for the reaction with 5 μL of FastStart probe master 2×, 1 μL of TaqMan^®^ probe 20×, 2.5 μL of gDNA at 20 ng/μL, and 1.5 μL of molecular biology grade water. gDNA at a final concentration of 50 ng was used. A total of 30% of the samples were analyzed by duplicated to avoid genotyping errors.

### 2.7. Dietary Intake 

We assessed the dietary intake with a 3-day dietary food record questionnaire (which included a day during the week, a weekend day, and a pre-intervention day). All subjects were instructed to provide the quantity and correct food description of their habitual dietary intake using a report of food consumption, for which participants were shown food scale models (Nasco^®^ Wisconsin, Fort Atkinson, WI, USA). The energy intake, and macro- and micronutrients composition were analyzed with the Nutritionist Pro™ version 8.1 software (Axxya Systems, Woodinville, WA, USA).

### 2.8. Breakfast Design

The breakfast fixed meal consisted of 2 slices of half-baked whole wheat (52 g), Hass avocado (58 g), natural turkey breast ham (48 g), tomato (20 g), grated carrot (60 g), 1 leaf of Italian lettuce (20 g), almonds (8 g), no sugar-added plain yogurt (114 g), Cantaloupe melon (160 g), and 236 mL of simple water at 22 °C. The ingredients used in the preparation of breakfast were of well-known commercial brands, carefully prepared, measured, and weighed by the staff of CUCSINE Food Service Management Laboratory (University Center of Health Sciences, University of Guadalajara, Guadalajara, Mexico). The energy composition of the breakfast was 526.5 kilocalories (43% carbohydrates, 21% protein, and 36% fat). Energy and macronutrient content were analyzed with the Nutritionist Pro™ version 8.1 software (Axxya Systems, Woodinville, WA, USA). 

### 2.9. Appetite Assessment 

Appetite was evaluated through visual analogue scales (VAS). The VAS are composed of a straight horizontal line of 100 mm with the words “None” or “Not at all” located at the left end and the words “Extreme” or “As much as I have never felt” at the right end. This instrument is frequently employed in appetite evaluation due to its reproducibility and simplicity [27]. Participants were asked to mark a transversal line with an ultra-fine point pen (Bic crystal, 0.7 mm), between the two ends of the scales, according to the appetite sensation (hunger, fullness, satiety, desire to eat, prospective food consumption) at the specific moment. The rate of such aspects of appetite was achieved by measuring the distance from the left end of the line to the mark and then, a numerical value was obtained. 

### 2.10. Statistical Analysis 

For the descriptive statistics, quantitative variables were expressed as mean ± standard deviation (SD) or median and interquartile range (IQRs); the qualitative variables were expressed as frequencies or percentages. The Shapiro–Wilk test was used to analyze the normality distribution of quantitative variables. Variables without normal distribution were log-transformed and their normality was checked again; the variables that maintained a non-normally distribution were analyzed with non-parametric statistical tests. The comparative analysis between two independent groups was analyzed with the Student’s *t*-test or the Mann–Whitney U test. The repeated measures analysis of variance (ANOVA) was used to analyze the scores of the VAS at different times and genotypes. Analysis of covariance (ANCOVA) test was used to analyze diet variables or hormone levels according to genotypes, adjusted by covariates and using Bonferroni correction for multiple comparisons. A *p*-value < 0.05 was considered statistically significant. All statistical analyses were performed using SPSS v28.0 software (IBM Corp., Armonk, NY, USA). Figures were made with GraphPad Prism v9.0 (GraphPad Software, San Diego, CA, USA) and with BioRender.com.

## 3. Results

### 3.1. Population Description

A total of 132 participants (77.3% women) were enrolled with a mean age of 22.0 ± 2.0 years. All anthropometric variables were different between women and men; however, no differences were found concerning the biochemical parameters, except in ghrelin levels, which were higher in men. In addition, women consumed more carbohydrates and less fat than men the night previous to the day of the intervention. Other variables such as dinner kilocalories the day before the intervention or the time of the last meal before intervention showed no differences between women and men (Table 1).

### 3.2. Genotypic and Allele Frequencies

All polymorphisms in this study were in Hardy–Weinberg Equilibrium (HWE). The genotype and allelic frequencies are shown in Table 2. The dominant model was used for comparisons of variables between genotypes; therefore, the frequencies of this model are reported in Table 2.

### 3.3. Dietary Intake According to Polymorphisms in GHRL and LEPR Genes

The dietary intake was self-reported, and no differences were found between the genotypes of *GHRL* or *LEPR* genes in energy intake or macronutrient intake (Appendix A). Nevertheless, the consumption of total sugars, fruit servings, and servings of bread/starch with added sugars were higher in subjects with the genotypes Leu/Met + Met/Met than in those with the Leu/Leu (Figure 3A,C,E). On the other hand, subjects with the Arg223 allele had a lower intake of total sugars and servings of bread/starch with added sugars in comparison with the Gln223Gln genotype (Figure 3B,F). Therefore, these variables were analyzed adjusting by the percentage of carbohydrates. Fruit servings per day and bread/starch with added sugar remain significant when a comparison was performed between Leu72Met genotypes, and total sugar was still significant between Gln223Arg genotypes (Table 3). The intake of dinner the day before of intervention was not different between Leu72Met genotypes, but higher kilocalories were consumed in subjects with the Gln223Gln genotype (Appendix A).

### 3.4. Subjective Appetite by Polymorphism Leu72Met of GHRL and Gln223Arg of LEPR Genes

Aspects of appetite (hunger, fullness, satiety, desire to eat, and prospective food consumption) were compared between genotypes of Leu72Met of *GHRL* and Gln223Arg of *LEPR* genes; however, no differences were found (Figure 4).

### 3.5. Appetite Hormones

The serum levels of the hormones of appetite were determined at the baseline and final time (120’ postprandial), and no differences were found between the Leu72Met genotypes. However, regarding the Gln223Arg polymorphism in the *LEPR* gene, the leptin basal levels were higher in carriers of wild-type genotype compared to those with the polymorphic allele (Table 4). Because of the differences observed in ghrelin levels between women and men (Table 1), as well as the differences observed in Figure 3 and Table 3, and previous reports related to body fat and ghrelin [28,29], levels of this hormone were analyzed between genotypes of Leu72Met, adjusting for such variables. In the same manner, due to results in Figure 3 and Table 3, and because it is well known that leptin levels are influenced by the amount of adipose tissue (which in turn differs by sex), comparison of leptin levels between Gln223Arg genotypes were also adjusted by sex, body fat percentage, and percentage of carbohydrates. It was observed that the levels of ghrelin at 120′ postprandial were higher in carries of Leu/Met + Met/Met compared with the Leu/Leu genotype (Figure 5). Other values showed no significant differences.

## 4. Discussion

Our study shows a novel association related to the genetic variants *GHRL* (Leu72Met) and *LEPR* (Gln223Arg) and dietary sugar intake, which provides a perspective that had not been previously reported. Additionally, we analyzed aspects of appetite, dietary intake, and appetite hormones according to genotypes of *GHRL* and *LEPR*. We found that Leu72Met and Gln223Arg polymorphism contribute to a differential response to the standardized meal by elevation of postprandial levels of ghrelin. When we applied the VAS for subjective appetite, we did not find differences related to the genotypes of *GHRL* and *LEPR*.

One interesting finding in this study was the higher fasting ghrelin concentrations in men compared to women. Similar results were found by Espinoza-García et al., who reported that in a group of young adults with normal weight, men presented higher concentration of ghrelin compared to women [30]. In contrast, Tobin et al. did not find differences in ghrelin fasting concentrations between men and women with overweight [31]. Studies have reported that differences in fasting ghrelin concentrations by sex may be related to the differences in energy intake, BMI, and body composition between men and women [32]. Indeed, we found differences between men and women in body composition, and carbohydrate and fat intake the night before ghrelin measurements.

The genotypic and allelic frequencies of the genotypes of *GHRL* Leu72Met in our study were similar to those reported in a Mexican-American population of Los Angeles [33]. Nevertheless, Rivera-León et al. observed different frequencies in a population of young adults from western Mexico with both normal weight and obesity; a possible explanation for this might be the use of different techniques to determine the genotypes [34]. Moreover, the allelic and genotypic frequencies of Gln223Arg genotypes of the *LEPR* gene were similar to those reported in the Mexican-American population of Los Angeles [35], as well as in young populations with normal weight and overweight in southern and western Mexico [34,35,36].

We compared dietary variables between genotypes, and we found differences in the intake of carbohydrates and sugars. To our knowledge, only the study of Takezawa et al. analyzed diet intake according to the Leu72Met polymorphism. They did not find significant differences in either the consumption of total energy, fat, or sugar in subjects with the Leu72Leu genotype, but the intake of dairy products was higher in subjects with the wild-type genotype [19]. Several studies have shown that meals with a high carbohydrate content achieve more effective suppression of ghrelin [36,37,38]; therefore, it is possible that subjects with at least one copy of the Met allele need to consume more carbohydrates to induce a successful suppressive effect. Indeed, it has been hypothesized that the effect of the polymorphism is a lower secretion and/or activity of this peptide [39]; consequently, these individuals would need a greater consumption of sugars, fruits, bread, etc., to achieve the same ghrelin suppressive effect after a meal as those without the Met variant.

Indeed, the higher postprandial ghrelin levels we found in the subjects with the Leu72Met polymorphism could be derived from the low-carbohydrate breakfast, as these individuals did not achieve a postprandial suppressive effect on ghrelin since, as mentioned above, they would need a higher carbohydrate intake to decrease the ghrelin levels to levels similar to those of Leu72Leu genotype. However, these results must be replicated in other populations that include more subjects with the Met allele and with different degrees of BMI. Ukkola et al. measured fasting ghrelin in a Swedish population with obesity and found that Met72Met carries have higher levels than subjects with the Leu72Leu or Leu72Met [40]. Additionally, Hedayatizadeh-Omran et al. compared the fasting total ghrelin serum levels for the study of coronary artery diseases in the Iranian population and found that the levels were higher in subjects with the polymorphic genotype Leu72Met + Met72Met [41]. However, we did not find information on ghrelin and the Leu72Met polymorphism of *GHRL* in the Mexican population. 

There are few studies about the Gln223Arg polymorphism in the *LEPR* gene and dietary intake, especially related to the servings of food groups. Dominguez-Reyes et al. observed that the consumption of carbohydrates was very similar between genotypes [42]. In addition, Mizuta et al. studied the preference for sweet food in relationship to the Gln223Arg polymorphism of *LEPR* in the Japanese adult population, in which no difference was reported regarding the Gln223Arg polymorphism [43]. Studies in murine models have shown that taste bud cells, mainly type II, express the ObR receptor and when leptin binds to it, the response to sweetness is suppressed [44,45]. However, another study has demonstrated the opposite: leptin increases the response of the chorda tympani nerve to the consumption of sucrose, in both fasted and free-fed mice [46]. Human studies have observed that in subjects without obesity, a synchronization was observed in the diurnal variations of leptin and sweet taste recognition thresholds, which suggests a connection between both systems [47]. The LEPR, TAS1R, and TAS3R receptors are encoded on chromosome 1 in humans [48,49], and polymorphisms in these genes may be in linkage disequilibrium. Therefore, the evaluation of the relationship between the consumption of simple sugars, leptin levels, sweet taste detection thresholds, and genetic variants will serve to clarify these mechanisms.

Concerning the subjective appetite assessment, we did not find previous studies that assess the association between these aspects of appetite and the two genetic variants of this study. Nevertheless, we did not find differences among genotypes, probably because the aspects of appetite do not always correlate with physiological variables. For example, ghrelin levels do not always correlate with hunger or leptin with satiety [50,51]. It is possible that these aspects of appetite do not depend on genetic traits but on the importance of other factors, especially psychological, behavioral, and environmental [52].

Limitations of this research include a small number of studied individuals and a different proportion of female and male participants. In addition, the lack of a control group during the intervention does not allow us to know how subjects with different genotypes respond to meals with different nutritional content. Finally, the application of eating behavior questionnaires and measuring short-term satiety hormones such as GLP-1 and PYY is recommendable.

## 5. Conclusions

The results of our study suggest that the genetic variables Leu72 Met of *GHRL* and Gln223Arg of *LEPR* could affect sugar intake in this population, since subjects with different genotypes of Leu72Met polymorphism responded differently to a standardized meal as evidenced by higher postprandial levels of ghrelin. This could lead to consuming more carbohydrates in the short- or medium-term (as we observed in the dietary intake between genotypes) and therefore being more susceptible to obesity at earlier ages. More studies of nutritional interventions considering this genetic variant are necessary.

## Figures and Tables

**Figure 1 nutrients-14-02100-f001:**
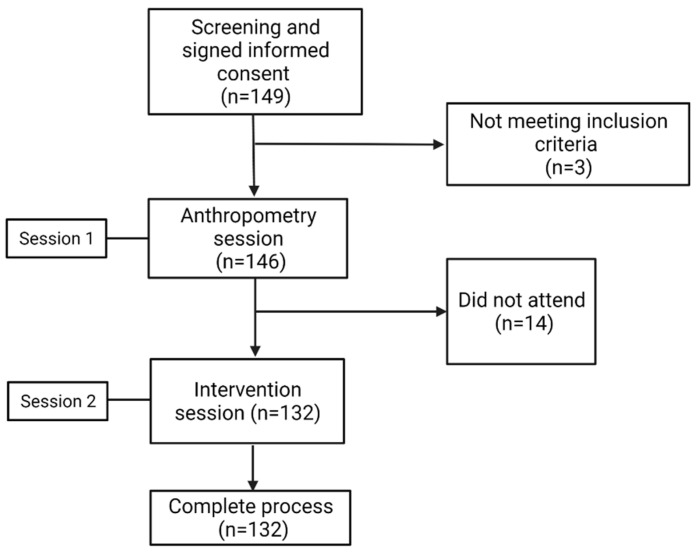
Flow diagram of subjects included in the study.

**Figure 2 nutrients-14-02100-f002:**
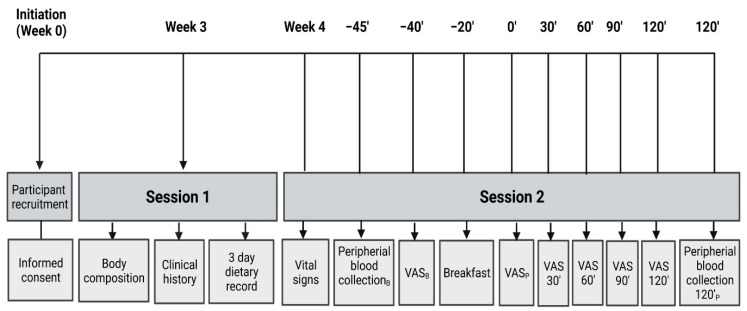
Schematic representation of the study procedure. VAS: Visual Analogue Scales; B: basal; P: postprandial.

**Figure 3 nutrients-14-02100-f003:**
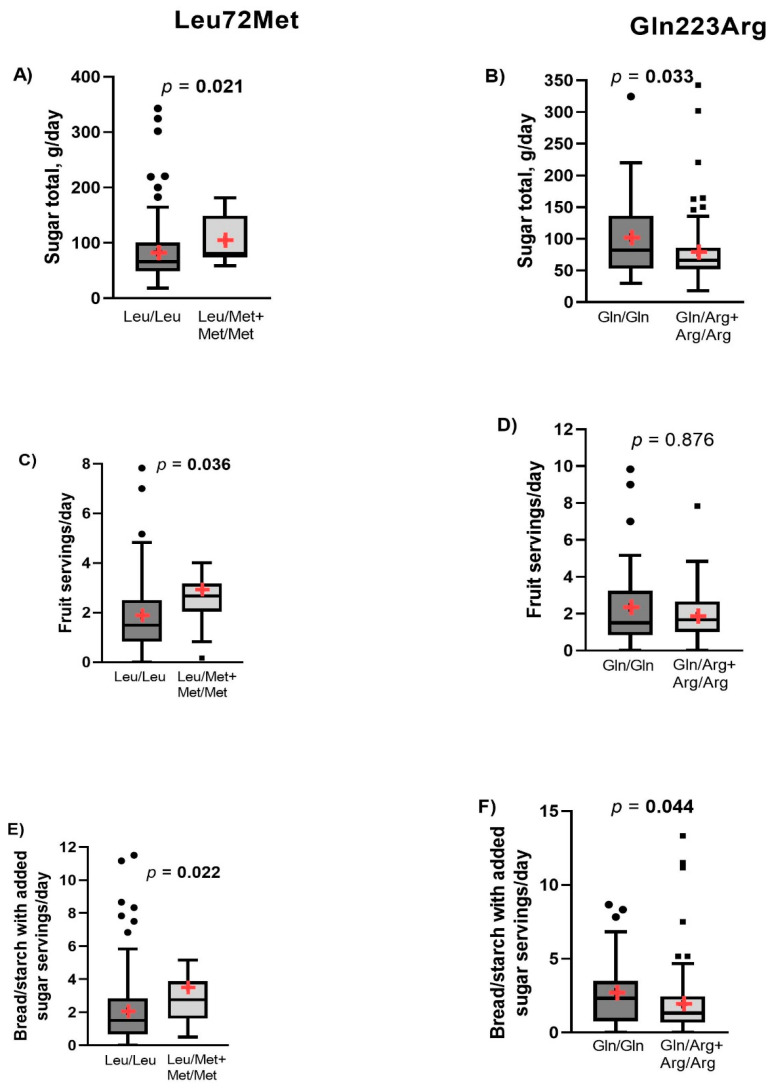
Differences in carbohydrate sources according to Leu72Met of *GHRL* and Gln223Arg of *LEPR*. Graphs (**A**,**C**,**E**) show dietary intake of total sugar, fruit servings, and bread/starch with added sugar servings between Leu72Met genotypes, respectively. Graphs (**B**,**D**,**E**) show the same dietary intake variables according to the Gln223Arg genotypes. The data are represented as median and IQR (percentile 25–75). The + symbol represents the mean. Differences between genotypes were calculated with the Mann–Whitney U test. A *p*-value < 0.05 was considered statistically significant. Bold numbers highlight statistical significance.

**Figure 4 nutrients-14-02100-f004:**
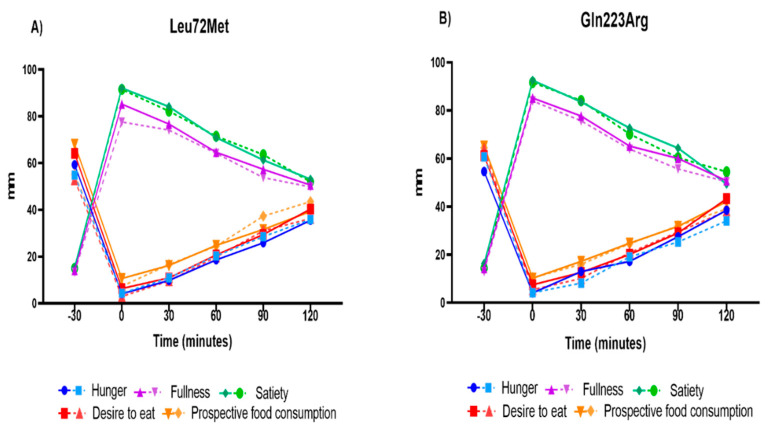
Subjective appetite by polymorphism Leu72Met of *GHRL* and Gln223Arg of *LEPR* genes. Graph (**A**) shows the comparison of the five subjective aspects of appetite between the Leu72Met genotypes. Graph (**B**) shows the comparison of the same variables between the Gln223Arg genotypes. A two-way repeated-measures ANOVA (analysis of variance) was used to compare time x genotype interaction for the subjective aspects of appetite. A *p*-value < 0.05 was considered statistically significant. The continuous line represents the homozygous genotype; the dashed line represents the heterozygous/homozygous polymorphic genotypes of Leu72Met of *GHRL* and Gln223Ag of *LEPR* genes, respectively.

**Figure 5 nutrients-14-02100-f005:**
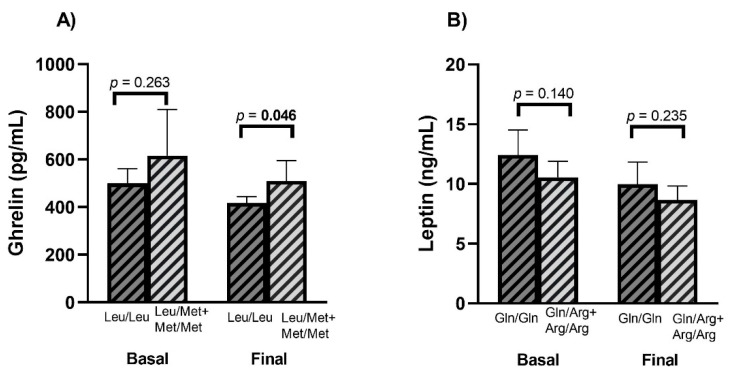
Adjusted appetite hormones levels according to Leu72Met of *GHRL* and Gln223Arg of *LEPR*. Graph (**A**) shows ghrelin concentrations between Leu72Met genotypes at fasting (basal) and at 120 minutes postprandial. Graph (**B**) shows leptin concentrations between Gln223Arg genotypes at fasting (basal) and at 120 minutes postprandial. Data are shown as estimated mean and 95% CI. ANCOVA (analysis of variance) test was used to compare appetite hormones between genotypes. Basal levels of ghrelin and leptin were adjusted by sex, body fat percentage, and percentage of carbohydrates, final levels of ghrelin and leptin (at 120 min postprandial) were adjusted by sex, body fat percentage, and percentage of carbohydrates, as well as for basal ghrelin and basal leptin, respectively. In all models, Bonferroni correction for multiple comparisons was used. A *p*-value < 0.05 was considered statistically significant. Bold numbers highlight statistical significance. CI: confidence interval.

**Table 1 nutrients-14-02100-t001:** General characteristics of the studied population.

Variables	All *n* = 132	Women *n* = 102	Men *n* = 30	*p*-Value
Anthropometric parameters				
BMI (kg/m^2^)	22.0 ± 2.0	21.7 ± 1.9	22.9 ± 2.1	**0.005**
WC (cm)	71.6 ± 6.4	69.6 ± 4.9	78.3 ± 6.2	**0.005**
Fat mass (kg)	17 ± 5.8	17.4 ± 4.5	16.2 ± 8.8	**<0.001**
BFP (%)	28.4 ± 7.1	30.4 ± 6.0	21.6 ± 6.4	**<0.001**
FFM (kg)	42.8 ± 8.3	39.5 ± 5.3	54.1 ± 6.7	**<0.001**
Lean mass (kg)	42.6 ± 28.8	40.1 ± 32.2	51.1 ± 6.3	**<0.001**
SMM (kg)	23.4 ± 5.1	21.3 ± 3.2	30.4 ± 4.0	**<0.001**
Mineral mass (kg)	3.0 ± 0.6	2.8 ± 0.4	3.7 ± 0.5	**<0.001**
TBW (kg)	31.3 ± 6.1	28.8 ± 3.8	39.7 ± 4.9	**<0.001**
Systolic blood pressure (mmHg)	109.4 ± 10.5	105.8 ± 8.3	121.5 ± 7.8	**<0.001**
Diastolic blood pressure (mmHg)	66.8 ± 7.2	66.0 ± 7.3	69.3 ± 6.5	**0.028**
Biochemical parameters				
TC (mg/dL)	146.7 ± 27.9	146.8 ± 28.2	146.3 ± 26.9	0.929
HDL-C (mg/dL)	49.7 ± 11.9	50.4 ± 11.8	46.8 ± 11.9	0.147
LDL-C (mg/dL)	80.3 ± 22.4	79.6 ± 23.0	82.4 ± 20.5	0.545
VLDL-C (mg/dL)	15.9 ± 5.9	15.8 ± 5.7	85.3 ± 34.2	0.294
Triglycerides (mg/dL)	80.0 ± 29.2	79.0 ± 28.2	46.8 ± 11.9	0.270
Glucose (mg/dL)	90.9 ± 11.9	90.4 ± 10.2	92.6 ± 10.7	0.366
Insulin (µUI/mL)	7.2 ± 4.2	7.4 ± 4.0	6.7 ± 4.8	0.395
HOMA-IR	1.6 ± 1.0	1.7 ± 1.0	1.5 ± 1.1	0.960
Ghrelin (pg/mL)	511.1 ± 346.5	466.5 ± 329.8	658.6 ± 364.8	**0.001**
Leptin (ng/mL)	11.1 ± 6.4	11.5 ± 6.6	10.0 ± 5.7	0.275
Others				
Kilocalories from dinner one day before intervention (kcal)	457.6 ± 301.1	448.2 ± 286.2	483.1 ± 344.3	0.674
Carbohydrates from dinner one day before intervention (%)	53.5 ± 19.1	56.2 ± 19.1	46.2 ± 17.5	**0.035**
Protein from dinner one day before intervention (%)	18.2 ± 9.7	17.7 ± 9.4	19.5 ± 10.3	0.476
Fat, total from dinner one day before intervention (%)	30.8 ± 14.5	28.8 ± 14.3	36.5 ± 13.7	0.031
Available carbohydrate from dinner one day before intervention (g)	1.1 ± 3.5	0.6 ± 2.8	2.5 ± 4.9	0.096
Alcohol from dinner one day before intervention (g)	0.02 ± 0.2	0.03 ± 0.2	0.0 ± 0.0	0.449
Dinner time one day before intervention (h)	21:23 ± 1:13	21:15 ± 1:13	21:3 ± 1:12	0.137

Data are shown as mean ± SD. Student’s *T*-test was used to compare variables between women and men. A *p*-value < 0.05 was considered statistically significant. Bold numbers highlight statistical significance. BMI: body mass index; WC: waist circumference; BFP: body fat percentage; FFM: fat-free mass; SMM: skeletal muscle mass; TBW: total body water; TC: total cholesterol; HDL-C: high-density lipoprotein cholesterol; LDL-C: low-density lipoprotein cholesterol; VLDL-C: very-low-density lipoprotein cholesterol; HOMA-IR: homeostatic model assessment-insulin resistance.

**Table 2 nutrients-14-02100-t002:** Genotype and allelic frequencies of Leu72Met of *GHRL* and Gln223Arg of *LEPR*.

**Leu72Met of *GHRL***		***n* (%)**
Genotype	Leu/Leu	120 (91)
	Leu/Met	12 (9)
	Met/Met	0 (0)
	HWE *p* = 0.5843	
Allele	Leu	252 (95)
	Met	12 (5)
Dominant model	Leu/Leu	120 (91)
	Leu/Met + Met/Met	12 (9)
**Gln223Arg of *LEPR***		***n* (%)**
Genotype	Gln/Gln	37 (28)
	Gln/Arg	68 (51)
	Arg/Arg	27 (21)
	HWE *p* = 0.8070	
Allele	Gln	142 (54)
	Arg	122 (46)
Dominant model	Gln/Gln	37 (28)
	Gln/Arg + Arg/Arg	95 (72)

HWE: Hardy–Weinberg equilibrium. *X*^2^ test was used to calculate the Hardy–Weinberg equilibrium.

**Table 3 nutrients-14-02100-t003:** Nutrient intake according to Leu72Met of *GHRL* and Gln223Arg of *LEPR*.

Variable	Leu/Leu *n* = 120	Leu/Met + Met/Met *n* = 12	*p*-Value	Gln/Gln *n* = 37	Gln/Arg + Arg/Arg *n* = 95	*p*-Value
Sugar total (g/d)	83.5 (74.6–92.4)	103.4 (75.4–131.4)	0.183	101.0 (85.2–116.9)	79.0 (68.9–89.0)	**0.021**
Fruit servings/d	1.9 (1.6–2.2)	2.9 (2.0–3.8)	**0.045**	2.3 (1.8–2.9)	1.9 (1.6–2.2)	0.166
Bread/starch with added sugar servings/d	2.1 (1.7–2.5)	3.5 (2.2–4.8)	**0.043**	2.7 (2.0–3.4)	1.9 (1.5–2.4)	0.086

Data are presented as estimated mean and 95% CI. ANCOVA (analysis of covariance) test was used to compare diet intake between genotypes, adjusted by the percentage of carbohydrates with Bonferroni correction for multiple comparisons. A *p*-value < 0.05 was considered statistically significant. Bold numbers highlight statistical significance. CI: confidence interval.

**Table 4 nutrients-14-02100-t004:** Serum levels of appetite hormones analyzed for association with Leu72Met of *GHRL* and Gln223Arg of *LEPR*.

Hormones	All *n* = 132	Leu/Leu *n* = 120	Leu/Met + Met/Met *n* = 12	*p*-Value	Gln/Gln *n* = 37	Gln/Arg + Arg/Arg *n* = 95	*p*-Value
Basal ghrelin (pg/mL)	511.1 ± 346.5 378.7 (317.9–653.5)	502.0 ± 340.4 377.3 (318.0–600.8)	600.2 ± 406.8 635.5 (287.5–726.8)	0.313	509.6 ± 420.4 366.0 (244.0–652.0)	514.4 ± 320.2 378.4 (330.0–692.5)	0.264
Final ghrelin (pg/mL)	420.9 ± 280.0 354.0 (272.8–475.0)	407.3 ± 264.6 350.0 (271.9–462.6)	556.1 ± 392.0 440.7 (347.2–701.6)	0.079	423.0 ± 394.8 321.7 (234.5–409.1)	419.0 ± 223.9 357.2 (276.3–494.7)	0.103
Basal leptin (ng/mL)	11.1 ± 6.3 10.8 (7.0–15.8)	11.0 ± 6.3 10.8 (7.0–15.0)	12.4 ± 7.8 11.3 (5.7–20.0)	0.534	12.4 ± 5.5 13.0 (8.0–17.0)	10.5 ± 6.6 10.0 (5.7–14.2)	**0.035**
Final leptin (ng/mL)	9.1± 6.4 7.9 (4.2–14.0)	9.1± 6.3 7.9 (4.2–14.0)	9.5 ± 7.9 7.0 (2.8–12.9)	0.889	10.4 ± 6.5 8.0 (5.0–14.0)	8.5 ± 6.3 6.3 (3.1–12.9)	0.085
Basal insulin (µUI/mL)	7.2 ± 4.2 6.1 (4.7–8.6)	7.2 ± 4.2 5.9 (4.5–8.4)	8.4 ± 4.5 6.9 (4.9–12.2)	0.379	7.5 ± 3.7 6.8 (4.9–9.1)	7.2 ± 4.5 6.0 (4.5–8.2)	0.403
Final insulin (µUI/mL)	12.6 ± 8.6 10.1 (6.7–15.9)	12.6 ± 8.8 10.0 (6.6–15.6)	12.6 ± 7.7 10.4 (7.2–20.7)	0.772	14.2 ± 12.0 9.4 (6.8–19.5)	11.9 ± 7.2 10.2 (6.5–15.1)	0.908

Data are shown as mean ± SD and median and IQR (percentile 25–75). Differences between genotypes were calculated with the Mann–Whitney U test. A *p*-value < 0.05 was considered statistically significant. Bold numbers highlight statistical significance. The final level refers to 120 min-postprandial levels.

## Data Availability

The data presented in this study are available on request from the corresponding author.

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
