# Peer review of "Role of Leu72Met of GHRL and Gln223Arg of LEPR Variants on Food Intake, Subjective Appetite, and Hunger-Satiety Hormones"

_nutrients, 2022, doi:10.3390/nu14102100_

Round 1

Reviewer 1 Report

Dear Authors,

The manuscript presents an interesting topic regarding the involvement of ghrelin and leptin SNPs in the aspects of human nutrition. Authors genotyped two SNPs mentioned above and found that both Leu72Met and Gln223Arg polymorphism contribute to differential response to the standardized meal by elevation of postprandial levels of ghrelin and higher total sugar intake. The study presents an important aspect of nutrition, confirming that genetic background plays a role in our response to food.

I have some minor considerations regarding missing elements in the manuscript.

  1. Please highlight the study's novelty at the begging of the discussion section.
  2. Please put the exact names of the statistical tests used to obtain particular results in the Tables' footnotes and the descriptions of the Figures, if appropriate.
  3. In figure 3, the + that represents the mean is hardly visible.
  4. There are some shortcuts without explanation in the manuscript. Please, explain the shortcuts in the proper order in the text.
  5. Line 317, if there is no statistically significant difference (p> 0.05), there are no "higher" or "lower" results.
  6. Line 363-364. Not including other aspects of appetite such as uncontrolled eating, disinhibition, or emotional eating is not a limitation of the study. Since such elements often accompany overweight or obese individuals, not those with normal BMI like the studied population.
  7. Line 226-227, please check the sentence's meaning; in line 299, please check the sentence.

8. The true limitation of the study is a relatively small number of studied individuals and also not an equal proportion of female and male participants. The authors should mention this in the limitation subsection.

Reviewer 2 Report

Comments see attachment.

Round 2

Reviewer 2 Report

Thank you for the revisions -- the manuscript is considerably more clear and much easier to follow.